# Secure Collaborative Computing for Linear Regression

Albert Guan [1,*], Chun-Hung Lin [2] and Po-Wen Chi [1]

1    Department of Computer Science and Information Engineering, National Taiwan Normal University, Taipei 11677, Taiwan; neokent@gapps.ntnu.edu.tw
2    Department of Computer Science and Engineering, National Sun Yat-sen University, Kaohsiung 80424, Taiwan; lin@cse.nsysu.edu.tw
*    Correspondence: albert.zj.guan@gmail.com

**Abstract:** Machine learning usually requires a large amount of training data to build useful models. We exploit the mathematical structure of linear regression to develop a secure and privacy-preserving method that allows multiple parties to collaboratively compute optimal model parameters without requiring the parties to share their raw data. The new approach also allows for efficient deletion of the data of users who want to leave the group and who wish to have their data deleted. Since the data remain confidential during both the learning and unlearning processes, data owners are more inclined to share the datasets they collect to improve the models, ultimately benefiting all participants. The proposed collaborative computation of linear regression models does not require a trusted third party, thereby avoiding the difficulty of building a robust trust system in the current Internet environment. The proposed scheme does not require encryption to keep the data secret, nor does it require the use of transformations to hide the real data. Instead, our scheme sends only the aggregated data to build a collaborative learning scheme. This makes the scheme more computationally efficient. Currently, almost all homomorphic encryption schemes that support both addition and multiplication operations demand significant computational resources and can only offer computational security. We prove that a malicious party lacks sufficient information to deduce the precise values of another party's original data, thereby preserving the privacy and security of the data exchanges. We also show that the new linear regression learning scheme can be updated incrementally. New datasets can be easily incorporated into the system, and specific data can be removed to refine the linear regression model without the need to recompute from the beginning.

**Keywords:** linear regression; machine learning; collaborative learning; privacy preserving; incremental computation; machine unlearning





## 1. Introduction

Machine learning attempts to build models based on training data to make predictions or decisions without being explicitly programmed. It usually requires a large amount of training data to build a useful model. In many practical applications, these data may not belong to one organization but rather may be distributed across multiple sites. For example, every hospital collects the medical data of its patients. The data collected in one hospital are often insufficient to obtain a good model for developing new drugs or treatments for some specific disease. Therefore, sharing data is very important in machine learning when the data are distributed across many sites.

Like medical data, almost all data contain private and sensitive personal information. The release of personal data in many countries in the world is restricted by laws and regulations. For example, since 2018, the European Data Protection Regulation has been applicable to all member states to harmonize data privacy laws across the European Union [1]. When sharing these data, security and/or personal privacy concerns must be properly addressed. Another reason for limiting data sharing is that collecting these data usually requires a lot of time and effort. Data owners usually view the data they collect as

an asset. They will be more willing to share the collected data with others only if it is legal and mutually beneficial.

In this paper, we consider the security and privacy preservation of a specific machine learning problem: linear regression. A linear regression models the relationship between variables by fitting a line to the observed data [2,3]. It is the most basic model in machine learning and has many applications [4]. We propose a simple yet effective method to collaboratively build linear regression models without disclosing the underlying raw data. Furthermore, our novel collaborative learning scheme enables the removal of a user's dataset without requiring access to the user's original data, ensuring compliance with privacy protection regulations.

Secure collaborative machine learning falls within the realm of secure multiparty computation, a well-explored area in the field of cryptography. In secure multiparty computing, no data can be sent directly to the other parties for security and privacy reasons. There are many techniques to achieve secrecy in multiparty computation. General method for secure multiparty computation, such as Yao's garbled circuits, may not be efficiently implementable. By exploiting the mathematical structure of linear regression, we achieve efficient secure construction of linear regression models.

The primary objective in secure computation is to establish a rigorous framework that ensures an absolute lack of knowledge concerning the private data of the other parties involved. Although achieving this state of zero knowledge is highly desirable, it typically necessitates extensive computational resources, as noted by Yang et al. [5]. Moreover, it is important to note that zero-knowledge guarantees are valid only against polynomial-time adversaries, meaning these schemes offer only computational security. The security of the proposed scheme does not depend on computationally hard problems, such as factoring large integers or solving discrete logarithms in a large group. We prove that the curious party does not have enough information to deduce the exact values of the other party's data. Therefore, even if the curious party has infinite computing power, he cannot learn the exact value of the other party's data.

A scheme with a trusted third party can be advantageous, particularly when the third party's role in the protocol is minimal. However, in today's Internet environment, establishing and maintaining long-term trust among multiple parties can be challenging, particularly as the number of participants grows. Such a system may lack robustness, as a breach in security by the server or any participating party can disrupt the system's functionality. The method proposed in this paper for collaboratively constructing a linear model operates without the need for trusted third parties. Our approach is based on the assumption that all parties are semi-honest, meaning they adhere to the protocol but maintain a curiosity about the data of other parties. This design ensures that the system remains functional even in the absence of unwavering trust among participants.

Encryption can be used to protect sensitive information in machine learning. However, in machine learning, data must be processed efficiently, and traditional encryption methods are not suitable for this purpose. Fully homomorphic encryption, such as the Paillier cryptosystem [6], has shown potential for computing model parameters in machine learning. Nonetheless, homomorphic encryption often requires substantial computational resources to achieve the required level of security. For example, the Halevi–Polyakov–Shoup fully homomorphic encryption scheme takes 51 ms for 128-bit security implemented by using a GPU [7]. Although this speed may be practical for cloud computing, the security of these cryptosystems is often based on computationally hard problems, so they can only provide computational security. The proposed scheme does not need encryption to protect users' raw data. We show that our proposed scheme can achieve information-theoretic security.

A critical consideration is that in machine learning, data frequently consist of real numbers, necessitating their representation in a standard format like IEEE Standard 754 floating-point numbers, which enables efficient computation using currently available computer hardware. To mitigate rounding errors, the plaintext and ciphertext of a cryptographic system is typically designed in finite groups, finite rings, or finite fields. Consequently,

the majority of these cryptographic systems, even when fully homomorphic, may not align seamlessly with the requirements of machine learning. To address this issue, recently, Cheon et al. developed the CKKS cryptosystem [8], which is designed to accommodate a specialized form of fixed-point arithmetic known as block floating-point arithmetic. While fixed-point arithmetic can approximate floating-point numbers within a finite memory size, it is worth noting that the current version of CKKS has limitations. Specifically, it supports only a limited number of multiplications before necessitating a time-consuming bootstrapping step. Another drawback of using the current version of CKKS is its lack of direct support for floating-point division operations, which are essential for tasks such as linear regression and for almost all machine learning models. Addressing this concern, Barbenko et al. proposed a two-stage method for Euclidean division within the CKKS scheme and conducted a study of its properties [9].

Datasets owned by a user may also be required to be efficiently deleted from computations. This is useful when a party leaves the group and wants to remove its information. This is called machine unlearning. Machine unlearning is necessary in order to comply with privacy protection laws and regulations, such as GDPR in European countries. In our proposed scheme, both the addition and deletion of datasets can be done efficiently without recomputing the model parameters from the beginning. However, our scheme needs to invert a matrix of size $(m + 1) \times (m + 1)$, where $m$ is the number of features used in the linear regression model. This is not a problem in practice, since the value of $m$ is usually small compared to the amount of data $n$.

The main contributions of this research are summarized as follows.

1.  We propose a secure and privacy-preserving machine learning scheme for linear regression without the requirement of trusted servers or encryption: notably, homomorphic encryption.
2.  The security of the proposed scheme does not depend on computationally hard problems. We prove that even if an attacker has unlimited computing power, he does not have enough information to compute the exact value of the other party's data.
3.  The proposed scheme is computationally efficient. It does not require encryption and does not incur additional computation due to distribution of the datasets.
4.  The proposed scheme is also communication efficient, as it only transmits aggregated data to other parties. The size of the aggregated data correlates with the number of features. In practical applications, whether data are encrypted or not, the number of data typically far surpasses the number of features.
5.  The proposed linear regression learning scheme supports incremental updates, allowing for the seamless integration of new datasets into the system.
6.  The proposed scheme also enables the removal of old datasets from the system, facilitating the refinement of the linear regression model without the need to recompute the model parameters from the beginning.

Organization of the paper. The remainder of this paper is organized as follows. In Section 2, we provide an overview of prior work in the field of federated learning; we focus on linear regression. We also presents a comparison of our scheme with other related schemes. In Section 3, we review linear regression, presenting it in matrix form and emphasizing the mathematical model that is the foundation for our approach. Our main results are described in Sections 4–6. In Section 4, we introduce our approach to privacy-preserving federated learning for linear regression. Section 5 is dedicated to the security analysis of our proposed scheme. Section 6 outlines techniques for implementing our scheme involving the continuous collection of data. In Section 7, we conclude our research work with key findings and contributions to the field of federated learning for linear regression. Finally, we describe our experience in implementing our proposed scheme in Appendix A. The results of these implementations are all consistent with our analysis.

## 2. Related Work

In contrast to traditional centralized machine learning techniques, for which all local datasets are uploaded to one server, the technique we study is known as federated learning or collaborative learning. In federated learning, the primary objective is to develop a method that enables parties to jointly compute the parameters of a machine learning model based on their input data while protecting the privacy of their individual data. To meet the security requirement, trusted servers can be used to protect the confidentiality of data and perform computations efficiently. The security of many known secure computation schemes for linear regression requires servers [10–15]. Some of the schemes use only one server [10,11,13]. Some of the schemes use two servers: one for cryptography service and the other for computation [12,14,15]. There are also different ways to use multiple servers. For example, in Cock et al.'s scheme [11], a trusted server is only needed in the pre-distribution phase to do initialization. On the other hand, in Chen et al.'s [10] and Mandal et al.'s schemes [13], the server is required to do computational tasks, but the server is only required to be semi-honest.

Federated learning can also be considered as secure multiparty computation or privacy-preserving computation. Secure multiparty computation has been widely studied by many cryptographers [16–18]. Recently, Ma et al. developed algorithms under secure multiparty computation that enable pharmaceutical institutions to achieve high-quality models to advance drug discovery without revealing private drug-related information [19].

The security of most existing secure multiparty computations depends on computationally hard problems, such as factoring large integers or solving discrete logarithms in a large group. Some of the schemes are based on Yao's garble circuit combined with secure computation for computing inner products [12]. Some of the schemes are based on homomorphic encryption to ensure the security of the data [14]. Kikuchi et al. proposed a scheme that does not require trusted servers [20], but it need Paillier's cryptosystem, which is a public-key homomorphic cryptosystem. Recently, Han et al. proposed two noninteractive federated learning schemes with privacy-preserving for IoMT [21], but they need trusted authority and Paillier's cryptosystem. These cryptosystems, including Yao's garbled circuits, are usually computationally intensive. Our proposed linear regression scheme operates without encryption, allowing us to circumvent the above issues and achieve efficient implementation.

Some of the known schemes do not use the closed form of the optimum solution for linear regression [12,13]. While the authors claim that the advantage is that their scheme does not require computing the matrix inverse, the disadvantage is that the result is only an approximation. On top of that, this computational method has to send gradient descent or other relevant data multiple times to make the model parameters accurate enough. It has been shown that sending gradient descent data iteratively can reveal enough information for the attacker to infer the values of the training data [22,23]. For each dataset, our scheme only sends the aggregated data once, and we show that the information leaked is too small for any unauthorized party to infer information about the other party's data. Note that this technique works well for linear regression problems. Other federated learning problems may not be able to avoid iterative sending of gradient descent data and may require other techniques to preserve privacy. For example, when using Deep Neural Networks (DNNs), Xu et al. proposed a double-masking protocol to guarantee the confidentiality of users' local gradients during federated learning [24].

Finally, we perform a comparative analysis of various federated learning schemes designed for linear regression. These schemes includes Cock's approach [11], Gascón's method [12], Kikuchi's model [20], Mandal's framework [13], Mohassel's system [15], Qiu's strategy [14], and Han's scheme [21]. These approaches, all relatively recent, are closely related to the security aspects of federated learning in the context of linear regression. All these schemes possess distinct characteristics that merit comparison, including whether they rely on a trusted server. Additionally, the choice of security technologies, such as homomorphic encryption or Yao's garbled circuit, plays a crucial role. Some schemes do

not employ closed-form solutions to attain optimal parameters but instead necessitate the iterative transmission of gradient descent updates for parameter refinement. Lastly, we evaluate the security levels of these schemes, with some achieving information-theoretic security while others rely on computational security. We take these factors into consideration in our comparative analysis, and the results of this comparison are presented in Table 1.

**Table 1.** Comparison of related protocols.

|  | Cock's [11] | Gascón's [12] | Kikuchi's [20] | Mandal's [13] | Mohassel's [15] | Qiu's [14] | Han's [21] | Ours |
|---|---|---|---|---|---|---|---|---|
| TA | Y | Y | N | Y | Y | Y | Y | N |
| HE | Y | Y | Y | Y | Y | Y | Y | N |
| YG(SMC) | N | Y | N | N | Y | Y | N | N |
| CF | Y | N | Y | N | N | N | N | Y |
| Security | C | C | C | C | C | C | C | P |

TA: needs trusted or semi-trusted authority. HE: uses homomorphic encryption. YG(SMC): uses Yao's garbled circuit for secure multiparty computation. CF uses closed-form formula to compute optimal parameters. Security: C means computationally secure; P means perfectly secure.

Note that schemes relying on trusted authorities (TAs) are difficult to implement in contemporary network environments due to trust issues. Homomorphic encryption (HE) and Yao's garbled circuit (YG(SMC)) are computationally intensive. When an iterative gradient descent is used instead of a closed form (CF), not only does it only produce approximate results, but it may also make the system vulnerable to data leakage. Finally, when quantum computers emerge, schemes for which the security relies on computationally hard problems may be easily broken.

## 3. A Brief Introduction to Linear Regression Model

In this section, we first briefly describe the computational procedure for linear regression. We then show a method to compute the optimal parameters of a linear model by minimizing the sum of the square of errors. This classical method is called the least square method [4,25].

Formally, linear regression is a linear approach to modeling the relationship between a response $y$ and a set of $m$ independent variables $x_1, x_2, \ldots, x_m$ in an experiment. It can be expressed as

$$y = \beta_0 + \beta_1 x_1 + \cdots + \beta_m x_m.$$

These independent variables $x_1, x_2, \ldots, x_m$ are also called the features of the model.

Suppose $n$ experiments have been performed, each with different settings for the features $x_1, x_2, \ldots, x_m$. With a set of $n$ observed data, we obtain $n$ equations.

$$y_i = \beta_0 + \beta_1 x_1^i + \beta_2 x_2^i + \cdots + \beta_m x_m^i, \quad i = 1, 2, \ldots, n,$$

where $x_j^i$ is the value of variable $x_j$ used in the $i$-th experiment. It is convenient to write these $n$ equations in matrix form. Let $\vec{x}_i = [x_1^i, x_2^i, \ldots, x_m^i]$, and let the model coefficients be: $\vec{\beta} = [\beta_0, \beta_1, \ldots, \beta_m]$. The model prediction would be

$$y_i = \beta_0 + \sum_{j=1}^{m} \beta_j x_j^i.$$

If each $\vec{x}_i$ is extended to $\vec{x}_i = [1, x_1^i, x_2^i, \ldots, x_m^i]$, then

$$y_i = \sum_{j=0}^{m} \beta_j x_j^i = \vec{\beta} \vec{x}_i^T.$$

In constructing a linear model, the coefficients $\beta_0, \beta_1, \ldots, \beta_m$ are unknowns. A set of $n$ experiments can be performed to determine possible values of these $m + 1$ unknowns. These $m + 1$ unknowns $\beta_0, \beta_1, \ldots, \beta_m$ are called parameters of the linear regression model.

If all the experiments have no errors and the setting of the variables $\vec{x}_i$ is independent, then $m + 1$ experiments are sufficient to determine the $m + 1$ unknowns $\beta_0, \beta_1, \ldots, \beta_m$. However, almost all experiments cannot avoid errors. More than $m + 1$ experiments—usually many more experiments—are needed to obtain "optimal" values of $\beta_0, \beta_1, \ldots, \beta_m$.

To define the optimal values for these parameters, we first define the loss function. Given a set of values for $\vec{\beta} = [\beta_0, \beta_1, \ldots, \beta_m]$, we can compute the loss with respect to $\vec{\beta}$ as

$$L(\vec{\beta}) = \sum_{i=1}^{n} \left( \vec{\beta} \vec{x}_i^T - y_i \right)^2.$$

In the least square setting, the optimum parameter is defined as the one that minimizes the sum of the mean square loss:

$$\hat{\vec{\beta}} = \arg \min L\left( \vec{\beta} \right) = \arg \min \sum_{i=1}^{n} \left( \vec{\beta} \vec{x}_i^T - y_i \right)^2.$$

## 4. The Proposed Collaborative Learning Method

In this section, we first describe the theoretical basis on which our proposed collaborative learning approach is based. We then describe new collaborative learning methods.

### 4.1. Theoretical Basis of the Collaborative Learning Method

First, we show how to compute the optimal parameters of a linear regression model incrementally. Assume that the number of features is two. Thus,

$$y = \beta_0 + \beta_1 x_1 + \beta_2 x_2.$$

The optimal values of each $\beta_j$, $j = 0, 1, 2$ can be obtained by finding the minimum value of the function $L(\vec{\beta})$. Let

$$L(\beta_0, \beta_1, \beta_2) = \sum_{i=1}^{n} (y_i - \beta_0 - \beta_1 x_1^i - \beta_2 x_2^i)^2.$$

Solve $\dfrac{\partial L}{\partial \beta_i} = 0$, $i = 0, 1, 2$. We obtain:

$$\sum_{i=1}^{n} (y_i - \beta_0 - \beta_1 x_1^i - \beta_2 x_2^i) = 0$$

$$\sum_{i=1}^{n} x_1^i (y_i - \beta_0 - \beta_1 x_1^i - \beta_2 x_2^i) = 0$$

$$\sum_{i=1}^{n} x_2^i (y_i - \beta_0 - \beta_1 x_1^i - \beta_2 x_2^i) = 0$$

These equations are also called normal equations. They can be rewritten as:

$$\sum_{i=1}^{n} y_i = \beta_0 n + \beta_1 \sum_{i=1}^{n} x_1^i + \beta_2 \sum_{i=1}^{n} x_2^i$$

$$\sum_{i=1}^{n} x_1^i y_i = \beta_0 \sum_{i=1}^{n} x_1^i + \beta_1 \sum_{i=1}^{n} (x_1^i)^2 + \beta_2 \sum_{i=1}^{n} x_1^i x_2^i \qquad (1)$$

$$\sum_{i=1}^{n} x_2^i y_i = \beta_0 \sum_{i=1}^{n} x_2^i + \beta_1 \sum_{i=1}^{n} x_1^i x_2^i + \beta_2 \sum_{i=1}^{n} (x_2^i)^2$$

The coefficients $\beta_0, \beta_1, \beta_2$ can be computed by solving systems of linear Equations (1).

Now suppose another set of $k$ data points are to be added:

$$(x_1^{n+1}, x_2^{n+1}, y_{n+1}), \ldots, (x_1^{n+k}, x_2^{n+k}, y_{n+k}).$$

Equations (1) become

$$\sum_{i=1}^{n+k} x_2^i = \beta_0(n+k) + \beta_1 \sum_{i=1}^{n+k} x_1^i + \beta_2 \sum_{i=1}^{n+k} x_2^i$$

$$\sum_{i=1}^{n+k} x_1^i x_2^i = \beta_0 \sum_{i=1}^{n+k} x_1^i + \beta_1 \sum_{i=1}^{n+k} (x_1^i)^2 + \beta_2 \sum_{i=1}^{n+k} x_1^i x_2^i$$

$$\sum_{i=1}^{n+k} x_2^i x_2^i = \beta_0 \sum_{i=1}^{n+k} x_2^i + \beta_1 \sum_{i=1}^{n+k} x_1^i x_2^i + \beta_2 \sum_{i=1}^{n+k} (x_2^i)^2$$

The new sum can be expressed as the old sum plus extra terms.

$$\sum_{i=1}^{n+k} x_2^i = \sum_{i=1}^{n} y_i + \sum_{i=n+1}^{n+k} x_2^i,$$

$$\sum_{i=1}^{n+k} x_1^i x_2^i = \sum_{i=1}^{n} x_1^i x_2^i + \sum_{i=n+1}^{n+k} x_1^i x_2^i,$$

$$\sum_{i=1}^{n+k} (x_1^i)^2 = \sum_{i=1}^{n} (x_1^i)^2 + \sum_{i=n+1}^{n+k} (x_1^i)^2$$

and so on for the other summations.

To generalize the method described above, it is more convenient to rewrite these equations by using matrices.

$$\begin{bmatrix} 1 & x_1^1 & x_2^1 & \ldots & x_m^1 \\ 1 & x_1^2 & x_2^2 & \ldots & x_m^2 \\ \vdots & \vdots & \vdots & \ddots & \vdots \\ 1 & x_1^n & x_2^n & \ldots & x_m^n \end{bmatrix} \begin{bmatrix} \beta_0 \\ \beta_1 \\ \vdots \\ \beta_m \end{bmatrix} = \begin{bmatrix} y_1 \\ y_2 \\ \vdots \\ y_n \end{bmatrix}.$$

The relationship between matrix $X$ and vector $\vec{y}$ can also be written in matrix form as:

$$X\vec{\beta} = \vec{y}. \tag{2}$$

The loss function can then be rewritten as:

$$\begin{aligned} L(\vec{\beta}) &= ||X\vec{\beta} - \vec{y}||^2 \\ &= \left(X\vec{\beta} - \vec{y}\right)^T \left(X\vec{\beta} - \vec{y}\right) \\ &= \vec{y}^T\vec{y} - \vec{y}^T X\vec{\beta} - \vec{\beta}^T X^T\vec{y} + \vec{\beta}^T X^T X\vec{\beta}. \end{aligned}$$

The optimum solution is at gradient 0, and the gradient of the loss function is:

$$\begin{aligned} \frac{\partial L(\vec{\beta})}{\partial \vec{\beta}} &= \frac{\partial \left(\vec{y}^T\vec{y} - \vec{y}^T X\vec{\beta} - \vec{\beta}^T X^T\vec{y} + \vec{\beta}^T X^T X\vec{\beta}\right)}{\partial \vec{\beta}} \\ &= -2X^T\vec{y} + 2X^T X\vec{\beta}. \end{aligned}$$

Setting the gradient to 0, we get the optimum coefficients:

$$-2X^T\vec{y} + 2X^TX\vec{\beta} = 0$$
$$\Rightarrow \quad X^T\vec{y} = X^TX\vec{\beta}$$
$$\Rightarrow \quad \vec{\beta} = \left(X^TX\right)^{-1}X^T\vec{y}.$$

Therefore, the optimum value for $\vec{\beta}$ is

$$\vec{\beta} = \left(X^TX\right)^{-1}X^T\vec{y}. \tag{3}$$

Suppose that each user $u$ has a set of $(x_1^i, x_2^i, \ldots, x_m^i)$ and its response $y_i$. They want to collaboratively compute the optimal coefficients

$$\vec{\beta} = (\beta_0, \beta_1, \ldots, \beta_m)$$

such that $||X\vec{\beta} - \vec{y}||^2$ is minimized. Assume that there are two parties: $A$ and $B$. $A$'s data are in the upper part of $X$ and $B$'s data are in the lower part of $X$. Matrix $X$ can be written as $\begin{bmatrix} X_1 \\ X_2 \end{bmatrix}$. Similarly, vector $\vec{y}$ can be written as $\begin{bmatrix} \vec{y}_1 \\ \vec{y}_2 \end{bmatrix}$. The system of Equation (2) can then be written as:

$$\begin{bmatrix} X_1 \\ X_2 \end{bmatrix}\vec{\beta} = \begin{bmatrix} \vec{y}_1 \\ \vec{y}_2 \end{bmatrix}$$

Applying Equation (3), the value of $\vec{\beta}$ can be computed as

$$\vec{\beta} = \left(\begin{bmatrix} X_1 \\ X_2 \end{bmatrix}^T \begin{bmatrix} X_1 \\ X_2 \end{bmatrix}\right)^{-1} \begin{bmatrix} X_1 \\ X_2 \end{bmatrix}^T \begin{bmatrix} \vec{y}_1 \\ \vec{y}_2 \end{bmatrix}$$

$$= \left(\begin{bmatrix} X_1^T & X_2^T \end{bmatrix} \begin{bmatrix} X_1 \\ X_2 \end{bmatrix}\right)^{-1} \begin{bmatrix} X_1^T\vec{y}_1 + X_2^T\vec{y}_2 \end{bmatrix}$$

$$= \left(\begin{bmatrix} X_1^TX_1 + X_2^TX_2 \end{bmatrix}\right)^{-1} \begin{bmatrix} X_1^T\vec{y}_1 + X_2^T\vec{y}_2 \end{bmatrix}$$

To compute the model parameter $\vec{\beta}$, the matrix needs to be inverted to become the (m+1) $\times$ $(m+1)$ matrix

$$\begin{bmatrix} X_1^TX_1 + X_2^TX_2 \end{bmatrix}.$$

In practical applications, the matrices that are to be maintained are

$$X^TX \text{ and } X^T\vec{y}.$$

These matrices are initialized as zero matrices. Whenever a new dataset $X_i$ and $\vec{y}_i$ are obtained, the new matrices $X_i^TX_i$ and $X_i^T\vec{y}_i$ are added to $X^TX$ and $X^T\vec{y}$, respectively. This enables the computation of a new set of model parameters for linear regression. It is also possible to delete a particular dataset, such as $X_j$ and $\vec{y}_j$, by subtracting $X_j^TX_j$ and $X_j^T\vec{y}_j$ from $X^TX$ and $X^T\vec{y}$, respectively.

### 4.2. Description of the Protocol

Suppose that $A$ and $B$ intend to utilize the least square method to construct a linear model that comprises $m$ features. Assume that $A$ has the data $X_1$ and the associated vector $\vec{y}_1$, and $B$ has the data $X_2$ and the associated vector $\vec{y}_2$. Our secure collaborative computation of the optimal parameter for linear regression is shown in Figure 1.

1. $A$ performs $n_1$ experiments and collects the data into an $n_1 \times (m+1)$ matrix $X_1$ and a $n_1 \times 1$ vector $\vec{y}_1$.
2. $B$ performs $n_2$ experiments and collects the data into an $n_2 \times (m+1)$ matrix $X_2$ and a $n_2 \times 1$ vector $\vec{y}_2$.
3. $A$ sends $X_1^T X_1$ and $X_1^T \vec{y}_1$ to $B$.
4. $B$ sends $X_2^T X_2$ and $X_2^T \vec{y}_2$ to $A$.
5. $A$ and $B$ can then compute the optimal model parameter $\vec{\beta}$, which is an $(m+1) \times 1$ vector:

$$\vec{\beta} = \left( \left[ X_1^T X_1 + X_2^T X_2 \right] \right)^{-1} \left[ X_1^T \vec{y}_1 + X_2^T \vec{y}_2 \cdot \right]$$

**Figure 1.** Secure collaborative learning of linear regression without encryption. In the protocol, assume that $A$ and $B$ intend to utilize the least square method to construct a linear model that comprises $m$ features.

## 5. Analysis of the Protocol

In this section, we first analyze the performance of the proposed protocol and then prove that our protocol is information-theoretically secure, assuming that all users are semi-honest, i.e., they follow the protocol but are curious to learn about the other parties' data.

### 5.1. Performance Analysis

Based on Section 4.1, it is easy to verify that our collaborative computation of the optimal parameter for linear regression is correct. For the computational complexity of the scheme, each party needs to compute $X_i^T X_i$ and $X_i^T \vec{y}_i$, $i = 1, 2$, respectively. This is necessary even if computations are performed by only one party using all the data in one set. The only possible extra work is that every party needs to invert the matrix $X_1^T X_1 + X_2^T X_2$ to obtain the solution of $\vec{\beta}$. The dimension of the matrix to be inverted is $(m+1) \times (m+1)$, where $m$ is the number of features used in the model, which is usually much smaller than the number of data $n$.

This step can also be done by one party (i.e., the server) who then sends the solution $\vec{\beta}$ to the other parties (i.e., local users). In this way, the total computational work is the same as the computational work required to be done when all the data are at one site.

For communication complexity, our proposed scheme sends only aggregated data to the other party. The size of the aggregated data is only $(m+1)^2 + (m+1) = (m+1)(m+2)$ numbers for each party having $n$ records of data. The communication complexity is proportional to the number of features $m$, which is independent of $n$. When $n > (m+1)$, this is less than the size of the dataset, which is $(n(m+1) + n)$.

### 5.2. Security Analysis

In this section, we show that the security of the proposed scheme is information-theoretically secure. This implies that even though a minor amount of information is inevitably divulged to establish a useful linear regression model, the attacker cannot compute the exact values of the other party's data, even if the attacker has infinite computing power.

We assume that all parties are semi-honest, i.e., the participants strictly follow the protocol, but they may be interested in knowing additional information that they are not explicitly permitted to know.

**Theorem 1.** *Assume that each party collects more than $m+1$ data. The proposed collaborative learning method for the optimal parameters of a linear regression model is secure in the following sense:*

1. *$A$ does not have enough information to compute any elements in $X_2$ and $\vec{y}_2$;*
2. *$B$ does not have enough information to compute any elements in $X_1$ and $\vec{y}_1$.*

**Proof.** Assume that some curious party attempts to calculate the values of the other party's data using the information he/she has collected in the protocol. We show that the number of unknown variables is greater than the number of equations he/she can formulate. This implies that the curious party cannot determine the exact value of the other party's data, even if they have infinite computing power.

Assume that $A$ has a set of $n_1$ data and $B$ has a set of $n_2$ data: $n_1 + n_2 = n$ and $n_1, n_2 > m + 1$, where $m$ is the number of features in the system. First we show that $A$ cannot compute the data of $B$ (i.e., elements of $X_2$ and $\vec{y}_2$) by using the information $A$ collects in the execution of the protocol.

The number of elements in $X_2$ is $n_2 \times (m + 1)$, but the first column of $X_2$ is a constant. Thus, there are $n_2 m$ variables. The number of elements in $\vec{y}_2$ is $n_2$, and it contains $n_2$ variables. Thus, $A$ has a total of $n_2(m + 1)$ unknowns.

On the other hand, the number of elements in $X_2^T X_2$ is $(m + 1)^2$, and the number of elements in $X_2^T \vec{y}_2$ is $m + 1$. The first element in the first row of $X_2^T \vec{y}_2$ is $n_2$. Thus, $A$ has only $(m + 1)^2 + n_2 - 1$ quadratic equations. Since $n_2 > m + 1$, $A$ does not have enough information to compute the values of the elements in $X_2$ and $\vec{y}_2$.

Similarly, we can show that $B$ does not have enough information to compute the values of the elements in $X_1$ and $\vec{y}_1$. □

Theorem 1 shows that even if the parties or the server have infinite computational power, there is not enough information to infer the values of the other party's data. That is, the security of the proposed linear regression scheme does not depend on computationally hard problems, such as factoring large integers or solving discrete logarithm problems in a large finite group. These computationally hard problems are solvable if the attacker has sufficient computing power: for example, by using a quantum computer.

Although we have proved that the curious party cannot compute the values of the dataset of the other party, even if they have infinite computing power, the curious party does know some information about the dataset. For example, the amount of data and the sum of the squares of the data, etc. For the information leakage in general, we note that Dwork and Naor show that if a machine learning model is useful, it must reveal some information about the data on which it was trained [26,27]. Our proof indicates that the information disclosed to the other party through the proposed collaborative linear regression scheme is minimal. As a result, none of the parties can leverage it to calculate the exact values of each other's data, even if the attacker has unlimited computing power.

As for how much information is lost, according to the proof of Theorem 1, user $A$ only has $(m + 1)^2 + n_2 - 1$ quadratic equations to solve $n_2(n + 1)$ unknowns. This means that $A$ will require

$$
\begin{aligned}
d &= n_2(m + 1) - (m + 1)^2 + n_2 - 1 \\
&= (m + 1)(n_2 - (m + 1)) + (n_2 - 1) > 0 \quad \text{if } n_2 > (m + 1)
\end{aligned}
$$

additional equations to solve for the value of each variable. These $d$ equations can be the values of any subset of the $d$ variables or of any system of $d$ equations on the variables. Without this additional information, there are $d$ free variables that can take on any value, and any possible values of these $d$ valuables can be used to calculate the values of other variables. Therefore, the information leaked is $(m + 1)^2 + n_2 - 1$ information units, where 1 information unit is equal to the value of the variable. Note that the total number of information units of the system is $n_2(n + 1)$, which is the total number of unknowns for the user $A$, or the uncertainty of $A$.

## 6. Practical Applications of the Scheme

The proposed collaborative learning method for linear regression can be viewed as a peer-to-peer mode of collaborative learning, or decentralized federated learning. In addition to the peer-to-peer model for collaborative learning, the proposed learning scheme can also be used for the client–server mode of collaborative learning, or centralized

federated learning. In the client–server mode of collaborative learning, the server may or may not own any training data. Each party sends its data to the server in the aggregated form. The server can then do all the computations for the users and can send the final result, i.e., the value of $\vec{\beta}$, to each user.

Users' new data can also be sent to the server many times. When sending their newly collected data incrementally, care must be taken to avoid sending a batch of too few data. For example, suppose that initially a user $A$ has collected a set of $n_1$ data. If $n_1 > m + 1$, then the user can send the corresponding matrix $G_1 = X_1^T X_1$ and $\vec{h}_1 = X_1^T \vec{y}_1$ to the server. Assume that more data were collected by the same user $A$. If the number of new data $n_2 > m + 1$, then $A$ can send another pair of matrices $G_2 = X_2^T X_2$ and $\vec{h}_2 = X_2 \vec{y}_2$ to the server. In the case that the number of data $n_2 \leq m + 1$, user $A$ may not want to send $G_2 = X_2^T X_2$ and $\vec{h}_2 = X_2^T \vec{y}_2$ to the server for security and privacy reasons. This is because the number of data in $X_2$ and $\vec{y}_2$ is too small, and a curious server can compute the matrix $X_2$ and $\vec{y}_2$ from $G_2$ and $\vec{h}_2$.

This implies that our proposed collaborate learning method can easily be implemented to compute optimal model parameters for linear regression incrementally. Suppose that $k$ datasets are collected over a period of time:

$$(X_1, \vec{y}_1), (X_2, \vec{y}_2), \cdots, (X_k, \vec{y}_k).$$

Define an $(m + 1) \times (m + 1)$ matrix

$$G_i = X_i^T X_i$$

and an $(m + 1) \times 1$ vector

$$\vec{h}_i = X_i^T \vec{y}_i.$$

The optimal model parameters $\vec{\beta}$ can be computed by the equation

$$\vec{\beta} = G^{-1} \vec{h},$$

where

$$G = \sum_{i=1}^{k} G_i, \text{ and } \vec{h} = \sum_{i=1}^{k} \vec{h}_i.$$

These $k$ sets of data $(X_1, \vec{y}_1), (X_2, \vec{y}_2), \cdots, (X_k, \vec{y}_k)$ can be the data of $k$ users collected at different times. When some dataset $(X_i, \vec{y}_i)$ is available, it can be added into $G$ and $\vec{h}$, respectively. Better model parameters $\vec{\beta}$ can then be computed.

It is also possible to delete some specified datasets as long as the relevant aggregated data are properly preserved. For example, if we want to delete all datasets associated with a party, we need to know the aggregated data for that party. Aggregated data of that party may also be held by that party. Once that party wants to delete his entire dataset, he can send the aggregated data to the server, which can deduct his data from the system.

By Theorem 1, in applying our scheme, it is required that the number of data $n_i$ must be greater than $m + 1$ for each user $i$. Otherwise, the curious party may be able to compute the value of the data set of user $i$. This is not a serious limitation, because, in any practical application, the number of features is usually small, and the amount of collected data should be much larger than the number of features to built an accurate machine learning model.

## 7. Conclusions and Discussion

We propose a simple and yet effective secure multiparty computation scheme for linear regression in a distributed environment. Our approach leverages the inherent mathematical properties of linear regression to allow participating parties to transmit only aggregated data to the server or other parties. This strategic approach enhances security and privacy protection without the need for encryption, especially homomorphic encryption. Security

and privacy play important roles in collaborative learning. By allowing data owners to protect their valuable data while still contributing to model improvement, our approach encourages greater data sharing and collaboration among participants.

Our proposed scheme offers the flexibility of incremental dataset management within the linear regression models. The seamless integration of new datasets into the system facilitates the acquisition of updated and more precise model parameters, all without the need for computation from the beginning. Additionally, the secure removal of user datasets is supported, which is required by laws and regulations when a party departs from the group and requests the deletion of his data.

The security of our proposed collaborated computation scheme for linear regression does not depend on computationally hard problems, such as factoring large integer and solving discrete logarithm in large finite group. These computationally hard problems usually require heavy computation to achieve a certain level of security. Furthermore, these computationally hard problems may be broken if the attacker has enough computing power, such as by using quantum computers. Due to the lack of sufficient information, we proved that the curious party cannot obtain exact values of the other party's data, even with unlimited computing power.

Our proposed method can also accommodate the inclusion of varying weights in collaborative learning. This feature holds particular promise in scenarios requiring customized machine learning, where all parties share commonalities but possess unique characteristics. A prime example is handwritten character recognition for smartphone input processing, for which personalized models are imperative. Our approach provides a customization process that enables users to build custom models by adding additional training datasets of their own to the generic model.

In the design of our linear regression protocol, we make no assumptions about how the dataset is collected and distributed within each site. In particular, the dataset collected at each site may contain different features. When applying our protocol, users must agree in advance on the features used in the model's calculations. Our approach belongs to horizontal federated learning as classified by Yang et al. citeYLCT19. In many applications, such as medical applications, the medical data collected by each hospital must contain a fixed set of common features that can be used to develop new drugs or treatments. Therefore, this is not a serious limitation.

**Author Contributions:** A.G.: contributed significantly to the analysis and manuscript preparation; C.-H.L.: contributed to the review and editing; P.-W.C.: contributed to the analysis with constructive discussions. All authors have read and agreed to the published version of the manuscript.

**Funding:** The research of the first author is supported in part by the MOST project 107-2218-E-110-017-MY3, and in part by NTNU special grant for new faculties. The research of the second author is supported in part by the MOST project 108-2622-E-110-009-CC3. The research of the third author is supported in part by the MOST project 110-2221-E-003-002-MY3.

**Informed Consent Statement:** Not applicable.

**Data Availability Statement:** Data is randomly generated and contained within the article.

**Acknowledgments:** We would like to thank Chao Wang for his contribution to the design of the experiments.

**Conflicts of Interest:** The authors declare no conflicts of interest.

## Appendix A. Implementation of the Proposed Scheme

In this appendix, we describe our experience for the implementation of our proposed privacy-preserving collaborative computing scheme for linear regression. We use the equation:

$$2 + x_1 - 2x_2 + 3x_3 + 2x_4 - x_5 + 2x_6 + 2.5x_7$$

as an example to show the correctness of our proposed scheme. The value of each feature $x_i$ in the experiment is randomly and uniformly selected from $[-50, 50]$. A Gaussian-distributed experimental error with mean $\mu = 0$ and variance $\sigma^2 = 5$ is added to $x_i$ in each experiment. The implementation environment is a PC equipped with an Intel CORE i7 CPU and running a Linux operating system with the gcc compiler.

*Appendix A.1. Incremental Computation for Linear Regression*

In this sub-section, we show that the incremental computation of linear regression of our scheme is correct.

Assume that a user $A$ first collected a set of $n_1$ data and then a set of $n_2$ data. These data are shown in Table A1:

**Table A1.** The fist dataset of user $A$ with $n_1 = 20$ experiments.

| $x_0$ | $x_1$ | $x_2$ | $x_3$ | $x_4$ | $x_5$ | $x_6$ | $x_7$ | $y$ |
|---|---|---|---|---|---|---|---|---|
| 1 | −16.26 | 18.44 | −22.2 | −46.48 | −32.63 | 26.94 | 19.2 | −73.2757 |
| 1 | 10.52 | 13.11 | −38.91 | 35.94 | 38.39 | 3.54 | −1.9 | −94.2417 |
| 1 | 45.59 | 38.25 | 18.11 | 30.93 | −39.72 | −49.24 | −26.39 | −38.4952 |
| 1 | −30.92 | −38.65 | −17.65 | 7.79 | −33 | −41.54 | −3.74 | −43.2171 |
| 1 | 11.72 | 0.15 | 16.01 | 8.98 | −31.41 | 7.34 | 12.5 | 156.469 |
| 1 | 49.49 | 47.8 | 45.23 | −26.47 | 10.91 | −43.68 | −40.52 | −163.418 |
| 1 | −0.69 | 9.86 | −28.9 | −5.1 | −38.37 | 39.21 | 39.36 | 102.719 |
| 1 | −28.09 | 3.5 | −37.03 | −9.01 | 14.85 | −41.16 | 48.78 | −137.248 |
| 1 | −4.63 | −32.7 | −41.44 | 20.61 | −19.03 | −11.9 | −20.41 | −81.5211 |
| 1 | −0.43 | 45.44 | 5.61 | −37.42 | 43.24 | 0.84 | −13.89 | −223.777 |
| 1 | 4.15 | −29.32 | −40.89 | 16.98 | −5.93 | −19.78 | 25.41 | 9.71093 |
| 1 | 5.7 | 32.95 | −21.71 | 27.62 | −13.55 | −8.74 | −31.39 | −153.078 |
| 1 | 14.82 | 0.1 | 30.92 | 23.71 | −19.08 | 3 | −5.68 | 171.14 |
| 1 | 11.9 | 41.1 | −12.57 | 24.99 | 0.06 | 43.05 | 37.57 | 125.301 |
| 1 | −6.7 | −6.11 | −26.32 | 10.98 | 14.58 | 46.32 | 41.48 | 130.289 |
| 1 | −41.35 | 40.06 | −19.59 | 14.35 | 23.01 | 8.7 | −44.51 | −266.432 |
| 1 | 22.99 | 49.96 | 37.63 | −48.67 | −36.42 | −17.93 | 25.05 | 5.34962 |
| 1 | −5.49 | 35.07 | 32.89 | −43.59 | −10.3 | −29.67 | 31.4 | −31.3214 |
| 1 | 39.76 | −36.62 | −17.51 | 46.59 | −29.21 | −30.31 | −28.91 | 49.7131 |
| 1 | −1.11 | −33.99 | −37.43 | 7.54 | −43.93 | −7.01 | 35.42 | 86.3319 |

In Table A1, as well as other data tables in this section, the last column is $y_i$.

The solution obtained by using the least square method on these 20 data is listed as follows. In the list of the solutions of the parameters, the number in the parentheses is the theoretical value.

$$\beta_0 = 2.48223\,(2)$$
$$\beta_1 = 0.943394\,(1)$$
$$\beta_2 = -1.99831\,(-2)$$
$$\beta_3 = 3.04756\,(3)$$
$$\beta_4 = 2.01561\,(2)$$
$$\beta_5 = -1.00936\,(-1)$$
$$\beta_6 = 1.99996\,(2)$$
$$\beta_7 = 2.53073\,(2.5)$$

Suppose that $A$ collected another set of data $A_2$ with $n_2 = 10$ data as shown in Table A2:

**Table A2.** The second dataset of user $A$ with $n_2 = 10$ experiments.

| $x_0$ | $x_1$ | $x_2$ | $x_3$ | $x_4$ | $x_5$ | $x_6$ | $x_7$ | $y$ |
|---|---|---|---|---|---|---|---|---|
| 1 | 10.65 | −47.49 | 9.25 | 12.57 | 9.24 | 35.2 | −1.66 | 217.627 |
| 1 | 10.88 | 14.43 | −2.51 | 45.02 | −9.74 | −19.05 | −37.63 | −56.4756 |
| 1 | −49.31 | 13.31 | −4.27 | 23.7 | −33.26 | 35.17 | −33.26 | −15.32 |
| 1 | −30.9 | −19.26 | 11.48 | 25.9 | 41.91 | −15.09 | 24.77 | 86.6076 |
| 1 | 47.06 | 0.63 | 24.81 | 7.71 | −33.33 | 47.58 | 33.8 | 352.092 |
| 1 | 25.91 | −3.7 | −17.86 | −49.69 | −39.27 | −6.85 | 8.85 | −68.6693 |
| 1 | −35.49 | 24.1 | 21.22 | −34.79 | −49.06 | −33.04 | 2.43 | −97.2314 |
| 1 | 31.2 | 15.65 | 19.18 | 13.82 | 46.4 | −19.34 | −10.27 | −24.3336 |
| 1 | 1.83 | −20.91 | 28.02 | −37.59 | −6.75 | 2.83 | 20.12 | 113.153 |
| 1 | 9.92 | 0.41 | 3.92 | 49.35 | 46.71 | −0.42 | 13.18 | 107.252 |

The solutions obtained by using the least square method on these 10 data is:

$$\beta_0 = 2.98339\,(2)$$
$$\beta_1 = 0.974087\,(1)$$
$$\beta_2 = -1.94836\,(-2)$$
$$\beta_3 = 2.92537\,(3)$$
$$\beta_4 = 2.01187\,(2)$$
$$\beta_5 = -1.00548\,(-1)$$
$$\beta_6 = 2.03572\,(2)$$
$$\beta_7 = 2.53478\,(2.5)$$

It can be seen that the solutions for each dataset are not accurate enough. These two datasets need to be combined to obtain a more accurate linear regression model. The two datasets can be computed incrementally by using our proposed scheme. The solutions

obtained by using the least square method on these two sets of $n = 20 + 10 = 30$ data incrementally is:

$$\beta_0 = 2.03899\,(2)$$
$$\beta_1 = 0.964677\,(1)$$
$$\beta_2 = -1.98436\,(-2)$$
$$\beta_3 = 3.01903\,(3)$$
$$\beta_4 = 2.01332\,(2)$$
$$\beta_5 = -1.01804\,(-1)$$
$$\beta_6 = 2.00631\,(2)$$
$$\beta_7 = 2.5225\,(2.5)$$

The model parameters are more accurate with the two sets.

*Appendix A.2. Collaborative Learning*

In this subsection, we demonstrate the proposed collaborative learning scheme with two users: *A* and *B*. We use the same equation:

$$2 + x_1 - 2x_2 + 3x_3 + 2x_4 - x_5 + 2x_6 + 2.5x_7$$

Furthermore, the environment in which the experiments are performed is the same as in the previous subsection.

Suppose that user *B* collected a dataset *B* with $n = 20$ data as given by Table A3:

**Table A3.** The dataset of *B* with $n = 20$ experiments.

| $x_0$ | $x_1$ | $x_2$ | $x_3$ | $x_4$ | $x_5$ | $x_6$ | $x_7$ | $y$ |
|---|---|---|---|---|---|---|---|---|
| 1 | −22.91 | −34.13 | 23.94 | 10.73 | −10.69 | −25.65 | 28.18 | 168.728 |
| 1 | −11.34 | −28.94 | −22.24 | −48.16 | 42.03 | −29.5 | 37.4 | −123.46 |
| 1 | −8.97 | 0.86 | −5.81 | −44.76 | −4.32 | −32.07 | −28.12 | −245.88 |
| 1 | 35.22 | −29.15 | −42.29 | 38.58 | 33.69 | 37.71 | −18.32 | 37.3495 |
| 1 | 11.89 | 10.33 | 2.79 | 38.98 | −10.28 | −23.26 | −36.77 | −49.6949 |
| 1 | 29.03 | −35.39 | 41.41 | 31.21 | 49.2 | −30.82 | −3.42 | 168.298 |
| 1 | 4.75 | −46.8 | −16.02 | 45.78 | 4.06 | 28.17 | −48.98 | 75.8072 |
| 1 | 13.26 | 9.63 | −27.1 | −38 | −6 | 44.13 | −49.42 | −193.578 |
| 1 | 41.22 | −4.63 | −17.74 | 3.11 | 19.22 | −1.42 | −44.39 | −130.139 |
| 1 | −41.05 | −11.16 | −31.16 | 1.5 | −33.03 | −39.74 | −17.28 | −190.617 |
| 1 | −33.83 | 42.96 | 29.3 | −15.56 | 46.16 | −36.72 | −19.78 | −231.838 |
| 1 | −36.26 | 4.98 | 44.76 | 27 | 28.13 | 31.19 | 39 | 274.162 |
| 1 | −27.87 | 25.32 | 39.58 | −36.65 | 34.21 | 35.37 | −20.02 | −44.4884 |
| 1 | 3.44 | 47.47 | −14.41 | −24.09 | 49.83 | −32.04 | 27.41 | −226.662 |
| 1 | −33.2 | −21.78 | −39.87 | −17.02 | 34.7 | 2.95 | 17.42 | −127.486 |
| 1 | 30.86 | −20.24 | 11.17 | 44.6 | 34.74 | −30.55 | −14.88 | 63.4341 |
| 1 | 12.87 | −49.36 | −25.88 | −1.48 | −10.51 | 27.22 | −24.6 | 35.2042 |
| 1 | −26.3 | −23.89 | 5.38 | −9.34 | 37.1 | 4.5 | −19.91 | −54.854 |
| 1 | 36.93 | 22.46 | −42.49 | −46.26 | 14.2 | 31.16 | −13.28 | −211.297 |
| 1 | −1.1 | 47.64 | 17.66 | 43.28 | −22.6 | 42.35 | 37.88 | 247.94 |

The solution obtained by using the least square method only on $B$'s dataset is:

$$\beta_0 = 1.70711 \,(2)$$
$$\beta_1 = 0.959823 \,(1)$$
$$\beta_2 = -1.99594 \,(-2)$$
$$\beta_3 = 2.98714 \,(3)$$
$$\beta_4 = 2.01443 \,(2)$$
$$\beta_5 = -1.02221 \,(-1)$$
$$\beta_6 = 1.98162 \,(2)$$
$$\beta_7 = 2.48759 \,(2.5)$$

If we use the proposed scheme to combine the two datasets collected by $A$ and $B$, we can obtain better model parameters:

$$\beta_0 = 2.01701 \,(2)$$
$$\beta_1 = 0.970765 \,(1)$$
$$\beta_2 = -1.9931 \,(-2)$$
$$\beta_3 = 3.00359 \,(3)$$
$$\beta_4 = 2.00549 \,(2)$$
$$\beta_5 = -1.02131 \,(-1)$$
$$\beta_6 = 1.99849 \,(2)$$
$$\beta_7 = 2.50664 \,(2.5)$$

(A1)

To verify the correctness of our proposed collaborative learning scheme, we also try to first combine the two datasets and then calculate the solutions. The results are:

$$\beta_0 = 2.01701 \,(2)$$
$$\beta_1 = 0.970765 \,(1)$$
$$\beta_2 = -1.9931 \,(-2)$$
$$\beta_3 = 3.00359 \,(3)$$
$$\beta_4 = 2.00549 \,(2)$$
$$\beta_5 = -1.02131 \,(-1)$$
$$\beta_6 = 1.99849 \,(2)$$
$$\beta_7 = 2.50664 \,(2.5)$$

(A2)

It can be seen that the solutions in Equations (A1) and (A2) are exactly the same.

*Appendix A.3. Performances of the Scheme*

First, we combine the $A$ dataset (30 data) and the $B$ dataset (20 data), and then we calculate the parameters of the linear regression model. The average computational times (in seconds) for each case executed 10 times are shown in Table A4:

**Table A4.** The computational times (in seconds). The two datasets are first combined.

|  | $m = 50$ | $m = 75$ | $m = 100$ |
|---|---|---|---|
| n = 10,000 | 0.133 s | 0.284 s | 0.511 s |
| n = 15,000 | 0.188 s | 0.411 s | 0.815 s |
| n = 20,000 | 0.300 s | 0.766 s | 1.467 s |

Then, we first compute the system parameters on the datasets of *A* and *B* separately and then consolidate the results. The time required to perform the calculations is listed below.

**Table A5.** The computational times (in seconds). The two datasets are not combined at first but are computed by using the proposed scheme.

|  | $m = 50$ | $m = 75$ | $m = 100$ |
|---|---|---|---|
| n = 10,000 | 0.123 s | 0.282 s | 0.512 s |
| n = 15,000 | 0.188 s | 0.442 s | 0.815 s |
| n = 20,000 | 0.298 s | 0.761 s | 1.456 s |

As can be seen from the above two tables, there is no obvious difference in the computational time based on either implementing our proposed scheme or combining the datasets first and then calculating the solutions. This is also consistent with our analysis that our scheme incurs no additional computational cost.

The above experiments do not include the communication time. The communication cost in our scheme is much lower because only $(m+1)(m+2)$ aggregated data needs to be exchanged, while the communication cost of those schemes that do not use aggregated data require the exchange of $n(m+2)$ data whether the data are encrypted or not.

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
