# Peer review of "Secure Collaborative Computing for Linear Regression"

_applsci, doi:10.3390/app14010227_

Round 1
Reviewer 1 Report (Previous Reviewer 2)
Comments and Suggestions for Authors
In this paper, the authors propose a new approach also allows to efficiently delete the data of users who wants to leave the group and wish to have their data deleted. In addition, the proposed collaborative computation of linear regression model does not require a trusted third party.
- This version has a much better than the previous version.
- - In Related Work, the authors may want to classify the related articles and introduce them in a comparative and conclusive way.
- - In section 6, line 365, the authors write “or security, in applying our scheme, it is required that the number of data ni must be greater than m + 1 for each user i”. The authors shout more explain this assumption.
Author Response
We would like to thank the reviewer for his/her valuable comments to improve the quality of the manuscript. We have revised the manuscript according to the comments of the reviewer. Major revisions are typed in red color in the revised manuscript. The responses to the comments of the reviewer are as follows.
1. This version has a much better than the previous version.
Reply. I am glad to hear that this version is considered an improvement over the previous one. Your feedback has been invaluable in shaping the manuscript.
2. In Related Work, the authors may want to classify the related articles and introduce them in a comparative and conclusive way.
Reply. We have rewritten parts of the Introduction section to make the new versionbetter organized. At the same time, some references and more comparisons have also been added to highlight the status and advantages of this article in this research field.
3. In section 6, line 365, the authors write “For security, in applying our scheme, it is required that the number of data ni must be greater than m + 1 for each user i”. The authors shout more explain this assumption.
Reply. The reason why this is needed is based on Theorem 1 proved in Section 5.7. We have revised the manuscript and state the reasons.
By Theorem 1, in applying our scheme, it is required that the number of data ni must be greater than m+ 1 for each user i. Otherwise the curious party may be able to compute the value of the data set of user i.
Reviewer 2 Report (New Reviewer)
Comments and Suggestions for Authors
1. The author needs to cite some recent works.
2. The results given by the author are not compared with existing work.
3. The abstract needs to be a little bit more precise and numerically conclusive.
4. The introduction's problem description is unclear in light of the paper's primary work, which does not address the problem description.
5. Rename heading number three as learning regression is a technique for mapping data linearly and producing predictions; it is not a problem.
6. In "The Linear Regression Problem" and "Theoretical Basis of the Proposed Scheme," the author uses the same phrases related to learning regression. Is it recommended that the author pass on their "Collaborative Learning Method" to learning regression?
Comments on the Quality of English LanguageOverall, an English translation is grammatically sound but the order of placement of topics often goes wrong and reading patterns get out of hand.
Author Response
We would like to thank the reviewer for his/her valuable comments to improve the qualityof the manuscript. We have revised the manuscript according to the comments of the reviewer. Major revisions are typed in red color in the revised manuscript. The responses to the comments of the reviewer are as follows.
1. The author needs to cite some recent works.
Reply. We have included some recent works, such as our recently discovered references.
[10] Cock, M.d.; Dowsley, R.; Nascimento, A.C.; Newman, S.C. Fast, Privacy
Preserving Linear Regression over Distributed Datasets Based on Pre-Distributed
Data. In Proceedings of the Proceedings of the 8th ACM Workshop on Artificial Intelligence and Security, New York, NY, USA, 2015; AISec ’15, pp. 3–14.
https://doi.org/10.1145/2808769.2808774.
[11] Gascon, A.; Schoppmann, P.; Balle, B.; Raykova, M.; Doerner, J.; Zahur, S.;
Evans, D. Privacy- Preserving Distributed Linear Regression on High-Dimensional
Data. Proceedings on Privacy Enhancing Technologies 2017, 2017, 345–364. https://doi.org/doi:10.1515/popets2017-0053.
[12] Mandal, K.; Gong, G. PrivFL: Practical Privacy-Preserving Federated Regressions on High- Dimensional Data over Mobile Networks. In Proceedings of the Proceedings of the 2019 ACM SIGSAC Conference on Cloud Computing Security Workshop, New York, NY, USA, 2019; CCSW’19, pp. 57–68. https://doi.org/10.1145/3338466.3358926.
[13] Qiu, G.; Gui, X.; Zhao, Y. Privacy-Preserving Linear Regression on Distributed
Data by Homomorphic Encryption and Data Masking. IEEE Access 2020, 8,
107601–107613.
https: //doi.org/10.1109/ACCESS.2020.3000764.
[14] Mohassel, P.; Zhang, Y. SecureML: A System for Scalable Privacy-Preserving
Machine Learning. In Proceedings of the 2017 IEEE Symposium on Security and
Privacy (SP), 2017, pp. 19–38.
https://doi.org/10.1109/SP.2017.12.
[19] Kikuchi, H.; Hamanaga, C.; Yasunaga, H.; Matsui, H.; Hashimoto, H.; Fan, C.I.
Privacy- Preserving Multiple Linear Regression of Vertically Partitioned Real Medical Datasets. Journal of Information Processing 2018, 26, 638–647. https://doi.org/10.2197/ipsjjip.26.638.
[20] Han, S.; Ding, H.; Zhao, S.; Ren, S.; Wang, Z.; Lin, J.; Zhou, S. Practical and Robust Federated Learning With Highly Scalable Regression Training.
IEEE Transactions on Neural Networks and Learning Systems 2023, pp. 1–15.
https://doi.org/10.1109/TNNLS.2023.3271859.
2. The results given by the author are not compared with existing work.
Reply. We have rewritten parts of the Introduction section. Some references and
more comparisons have been added to highlight the status and advantages of this article in this research field. A more complete comparisons was given at the end of Related Work section.
3. The abstract needs to be a little bit more precise and numerically conclusive.
Reply. We have rewritten parts of the Abstract section to make the new version better more precise and more conclusive.
4. The introduction’s problem description is unclear in light of the paper’s primarywork, which does not address the problem description.
Reply. We have rewritten parts of the Introduction section to make the new version better organized, more clear and better organized to show the importance of the present work.
5. Rename heading number three as learning regression is a technique for mapping data linearly and producing predictions; it is not a problem.
Reply. We renamed this section to “A Brief Introduction to Linear Regression Model”.
6. In ”The Linear Regression Problem” and ”Theoretical Basis of the Proposed Scheme,” the author uses the same phrases related to learning regression. Is it recommended that the author pass on their ”Collaborative Learning Method” to learning regression?
Reply. We rename this section “Theoretical Foundations of Collaborative Learning Methods”.
Reviewer 3 Report (New Reviewer)
Comments and Suggestions for Authors
The article proposes a new method for secure and privacy-preserving machine learning for linear regression without the requirement of trusted servers or encryption. The proposed scheme is computationally efficient and communication efficient, as it only transmits aggregated data to other parties. The proposed linear regression learning scheme supports incremental updates, allowing for the seamless integration of new datasets into the system. The proposed scheme also enables the removal of old datasets from the system, facilitating the refinement of the linear regression model without the need to recompute the model parameters from the beginning. The security of the proposed scheme does not depend on computationally hard problems, such as factoring large integers or solving discrete logarithms in a large group. The proposed scheme operates without encryption, allowing us to circumvent the above issues and achieve efficient implementation. The article also provides an overview of prior work in the field of federated learning, focusing on linear regression, and presents a comparison of the proposed scheme with other related schemes.
One minor notation issue is found. As vectors x_i and beta are both row vectors, one of which needs to be transposed before the multiplication.
The authors should also discuss the limitation of the application, whether the method is suitable for all types of datasets.
Author Response
We would like to thank the reviewer for his/her valuable comments to improve the quality of the manuscript. We have revised the manuscript according to the comments of the reviewer. Major revisions are typed in red color in the revised manuscript. The responses to the comments of the reviewer are as follows.
1. One minor notation issue is found. As vectors xi and β are both row vectors, one of which needs to be transposed before the multiplication.
Reply. We have revised the typo.
2. The authors should also discuss the limitation of the application, whether the method is suitable for all types of datasets.
Reply. We have discuss the limitation of our protocol in the application. The following paragraph has been added at the end of the Conclusion and Discussion section.
In the design of our linear regression protocol, we make no assumptions about how the dataset is collected and distributed within each site. In particular, the dataset collected at each site may contain different features. When applying our protocol, they must agree in advance on the feature used in the model computation. Our approach belongs to the horizontal federated learning as classified by Yang et al. [5]. In many applications, such as medical applications, the medical data collected by each hospital must contain a fixed set of common features that can be used to develop new drugs or treatments. Therefore, it is not a serious limitation.
Round 2
Reviewer 2 Report (New Reviewer)
Comments and Suggestions for Authors
this paper is technically incomplete because the result section of this paper is missing.
Comments on the Quality of English Languageoverall english in this paper is ok but some times breaking the flow of reading.
Author Response
We would like to thank the reviewer for his/her valuable comments to improve the quality of the manuscript. We have revised the manuscript according to the comments of the reviewer. Major revisions are typed in red color in the revised manuscript. The responses to the comments of the reviewer are as follows.
1. This paper is technically incomplete because the result section of this paper is missing.
Reply. The main results are described in the protocol section, the security analysis section, and the application section. We rewrite the organizational part of the Introduction section as follows.
Organization of the paper. The remainder of this paper is organized as follows: In Section 2 we provide an overview of prior work in the field of federated learning, focusing on linear regression. We also presents a comparison of our scheme with other related schemes. In Section 3 we review linear regression, presenting it in matrix form and emphasizing the mathematical model that are the foundation for our approach. Our main results are described in Section 4, Section 5, and Section 6. In Section 4 we introduce our approach to privacy-preserving federated learning on linear regression. Section 5 is dedicated to the security analysis of our proposed scheme. Section 6 outlines techniques for implementing our scheme involving the continuous collection of data. In Section 7, we conclude our research work with key findings and contributions to the field of federated learning for linear regression. Finally, we describe our experience in implementing our proposed scheme in Appendix Section. The results of these implementations are all consistent with our analysis.
2. Overall english in this paper is ok but some times breaking the flow of reading.
Reply. We have rewritten some passages of the paper to make it flow more smoothly. Major revisions are highlighted in red at the end of Section 2. We also made other rewrites to make the paper more readable.
This manuscript is a resubmission of an earlier submission. The following is a list of the peer review reports and author responses from that submission.
Round 1
Reviewer 1 Report
Comments and Suggestions for Authors
The paper discusses an interesting idea, albeit not new. However, the main flaw in this article is the lack of experimental evidence.
- Throughout the results' discussion and conclusion sections, the authors have provided many bold unjustified claims.
- Unless accompanied by an experimental setup to test the system and provide evidence for the superiority claims, it is still a paper presenting a theoretical idea that has been borrowed from federated learning.
- The case study provided is a merely repeated discussion to allow readers to infer the conclusions stated. However, these claims require experimental proof.
- The qualitative comparison given in Table 1 is not sufficient. You need to compare computational power and time consumption and immunity against attacks as well to justify your conclusions drawn.
- Attack risks on security during the collaboration process are presumably reduced. Yet there is no evidence provided to demonstrate the ideas presented. One can only infer a reduced load when sharing only the features learned by each party. Other than that, I find it difficult to convince readers of the benefit of such a scheme that lacks encryption and competes against federated learning schemes.
- The presentation is somewhat cut abruptly, it gives the feeling that the authors would like to publish the results in a different venue. If that is the case, perhaps this paper should go to a conference venue first.
Author Response
We would like to thank the reviewer for his/her valuable comments to improve the quality of the manuscript. We have revised the manuscript according to the comments of the reviewer. Major revisions are typed in red color in the revised manuscript. The responses to the comments of the reviewer are as follows.
1. The paper discusses an interesting idea, albeit not new. However, the main flaw in this article is the lack of experimental evidence.
Reply. Almost all current approaches to solving collaborative learning in linear regression use encryption and/or trusted center. Using encryption requires computation time, homomorphic encryption takes even longer, and currently most homomorphic encryption does not directly provide division operations for floating point numbers.
By analyzing the mathematical structure of linear regression, we propose a simple and effective method to solve collaborative learning of linear regression problems without encryption and trusted centers. We have analyzed the correctness and performance of the proposed protocol. The results of any implementation of the protocol should be consistent with the theoretical analysis. Therefore, we did not include experimental results in the manuscript.
2. Throughout the results’ discussion and conclusion sections, the authors have provided many bold unjustified claims.
Reply. We have analyzed the correctness and performance of the proposed protocol. We have also rewritten many parts of the manuscript. The revised version of the manuscript should make it easier for readers to understand the reasoning for the conclusions.
3. Unless accompanied by an experimental setup to test the system and provide evidence for the superiority claims, it is still a paper presenting a theoretical idea that has been borrowed from federated learning.
Reply. In addition to the theoretical importance, we believe our findings also have practical implications. Encryption takes computation time, homomorphic encryption takes even longer to complete. Most importantly, like any public key cryptosystems, homomorphic encryption can only provide computational security. We havshown that our scheme can achieve information-theoretical security without trusted third party. That is, even if an attacker has unlimited computing power, he does not have enough information to deduce the exact value of the other party’s data.
4. The case study provided is a merely repeated discussion to allow readers to infer the conclusions stated. However, these claims require experimental proof.
Reply. We have rewritten many parts of the manuscript in the hope that readers
will be easier to understand the reasons why conclusions can be drawn from the
theorems we have proved.
5. The qualitative comparison given in Table 1 is not sufficient. You need to compare computational power and time consumption and immunity against attacks as well to justify your conclusions drawn.
Reply. It is known that encryption requires computational time, and using public key encryption takes 100 to 1000 times longer than symmetric key encryption.
Homomorphic encryption takes longer than public key encryption which does not offer homomorphic properties. Since our proposed scheme does not require any encryption, it is clear that our scheme runs faster than those which need encryption. Therefore, comparing the schemes with the properties in the current table reveals information we need to know, and justify conclusions we drawn.
6. Attack risks on security during the collaboration process are presumably reduced. Yet there is no evidence provided to demonstrate the ideas presented. One can only infer a reduced load when sharing only the features learned by each party. Other than that, I find it difficult to convince readers of the benefit of such a scheme that lacks encryption and competes against federated learning schemes.
Reply. We have stated that the security model in our scheme is that all parties are semi-honest. That is, they all follow the protocol but are curious about knowing the other party’s data. This is a basic security model for collaborative learning. The presence of malicious parties would be a good topic for future research.
7. The presentation is somewhat cut abruptly, it gives the feeling that the authorswould like to publish the results in a different venue. If that is the case, perhaps this paper should go to a conference venue first.
Reply. We have rewritten many parts of the manuscript. The major revisions are
typed in red color in the revised manuscript. This article has been submitted for
publication in this journal for publication and is not intended for submission to any other venue.
Reviewer 2 Report
Comments and Suggestions for Authors
In this paper, the authors propose a new approach also allows to efficiently delete the data of users who wants to leave the group and wish to have their data deleted. In addition, the proposed collaborative computation of linear regression model does not require a trusted third party. The structure of this paper is well organized and explanation of key part is clear, thus the paper is easy to understand. However, the reviewer has some comments, as is listed below.
- The abstract can be improved.
- I suggest the authors add their paper's structure at the end of “Introduction”.
- In section 4, the authors present the proof of theorem 1 for collaborative learning method for the optimal parameters of a linear regression model. However, the authors should make more efforts to explain this proof more.
Author Response
We would like to thank the reviewer for his/her valuable comments to improve the quality of the manuscript. We have revised the manuscript according to the comments of the reviewer. Major revisions are typed in red color in the revised manuscript. The responses to the comments of the reviewer are as follows.
In this paper, the authors propose a new approach also allows to efficiently delete the data of users who wants to leave the group and wish to have their data deleted. In addition, the proposed collaborative computation of linear regression model does not require a trusted third party. The structure of this paper is well organized and explanation of key part is clear, thus the paper is easy to understand. However, the reviewer has some comments, as is listed below.
1. The abstract can be improved.
Reply. We have rewritten the Abstract Section. The modifications are typed in red color.
2. I suggest the authors add their paper’s structure at the end of “Introduction”.
Reply. A structure of the manuscript has been added at the end of Introduction
Section.
3. In section 4, the authors present the proof of theorem 1 for collaborative learning method for the optimal parameters of a linear regression model. However, the authors should make more efforts to explain this proof more.
Reply. We have added the technique of the proof at the beginning of the proof. The technique of proof can help readers understand how proofs work.
Reviewer 3 Report
Comments and Suggestions for Authors
The paper presents a secure multi-party computation scheme for linear regression in a federated learning environment. It argues that the proposed method, which only requires the sending of aggregated data between parties, offers better security and privacy without relying on encryption or computationally hard problems.
Strengths
A comparison is made with existing schems, pointing out the advantages of the proposed method in terms of computational load and security. This provides valuable context for the reader.
The paper outlines how the proposed scheme can be implemented in both peer-to-peer and client-server models, making it flexible for various types of federated learning architectures.
The paper does offer significant contributions to the field of federated learning and secure multi-party computation, particularly in terms of its analysis, comprehensive comparison with existing schemes, and practical implications.
Weaknesses
The paper assumes that all participating parties are semi-honest, which might not hold true in real-world applications.
Although it is mentioned that parties may collect data with a different set of features, the paper does not delve into how this would impact the protocol. I suggest that the authors clarify the part about different sets of features among different parties and how it would affect the system.
While the paper claims information-theoretical security, it acknowledges that some information leakage is inevitable. However, it does not provide a quantification of this leakage. Please provide quantification or metrics for the inevitable information leakage mentioned in the security analysis.
Comments on the Quality of English Language
Typos and grammatical issues:
The paper contains several typos and grammatical issues. For instance, there's a typo in the title of Section 4.5. Comparsions, it should be Comparisons. Also, in the same section, the phrase "unrealistic and unrealistic" appears to be a typo. Proofread the manuscript carefully to eliminate typos and grammatical errors.
Author Response
We would like to thank the reviewer for his/her valuable comments to improve the quality of the manuscript. We have revised the manuscript according to the comments of the reviewer. Major revisions are typed in red color in the revised manuscript. The responses to the comments of the reviewer are as follows.
General
The paper presents a secure multi-party computation scheme for linear regression in a federated learning environment. It argues that the proposed method, which only requires the sending of aggregated data between parties, offers better security and privacy without relying on encryption or computationally hard problems.
Strength
1. A comparison is made with existing schemes, pointing out the advantages of the proposed method in terms of computational load and security. This provides valuable context for the reader.
2. The paper outlines how the proposed scheme can be implemented in both peerto-peer and client-server models, making it flexible for various types of federated learning architectures.
3. The paper does offer significant contributions to the field of federated learning and secure multi-party computation, particularly in terms of its analysis, comprehensive comparison with existing schemes, and practical implications.
Reply. Thanks to the reviewer for the approval.
Weaknesses
1. The paper assumes that all participating parties are semi-honest, which might not hold true in real-world applications.
Reply. The current manuscript deal with semi-honest model. In many applications, if the privacy of the data can be preserved, participants are willing to share their data to build more accurate models, thus benefiting all participants. Therefore, it is feasible to assume that all participants are semi-honest, especially when data privacy is required by law. However, the presence of malicious parties would be an excellent topic for future research.
2. Although it is mentioned that parties may collect data with a different set of features, the paper does not delve into how this would impact the protocol. I suggest that the authors clarify the part about different sets of features among different parties and how it would affect the system.
Reply. The current version of the manuscript handles a fixed set of features for all participants. If the data collected by each participant has different features, then they must agree on which features to be used first. The selected features must be in the subset of the intersection of all features of the data. This limitation may not be a big problem in practical applications. For example, to develop a model for medical treatment, the data collected by hospitals must contain some common features which is determined based on medical research. We have stated this fact in Section 7, page 11.
3. While the paper claims information-theoretical security, it acknowledges that some information leakage is inevitable. However, it does not provide a quantification of this leakage. Please provide quantification or metrics for the inevitable information leakage mentioned in the security analysis.
Reply. We have estimated the amount of information leaked in Section 5, page 9.
Typos and grammatical issues:
4. The paper contains several typos and grammatical issues. For instance, there’s a typo in the title of Section 4.5. Comparsons, it should be Comparisons. Also,
in the same section, the phrase ”unrealistic and unrealistic” appears to be a typo.
Proofread the manuscript carefully to eliminate typos and grammatical errors.
Reply. We have corrected these typos and errors.
Round 2
Reviewer 1 Report
Comments and Suggestions for Authors
The authors have still provided no experimental evidence to their claims. Hence, my decision did not change.
I recommend Reject for lacks of supporting experimental evidence.
Comments on the Quality of English LanguageNo news results were provided in the paper No change in my decision
Author Response
We would like to thank the reviewer for his/her valuable comments to improve the quality of the manuscript. We have revised the manuscript according to the comments of the reviewer. Major revisions are typed in red color in the revised manuscript. The responses to the comments of the reviewer are as follows.
1. The authors have still provided no experimental evidence to their claims.
Reply. We have done several experiments and the results show that they are consistent in our claim. We have added the experimental section in Section 7.
Reviewer 2 Report
Comments and Suggestions for Authors
All my comments have been addressed.
Author Response
We would like to thank the reviewer for his/her valuable comments to improve the quality of the manuscript. We have revised the manuscript according to the comments of the reviewer. Major revisions are typed in red color in the revised manuscript. The responses to the comments of the reviewer are as follows.
1. All my comments have been addressed.
Reply. Thanks.